# Novel Derivatives of Quinoxaline-2-carboxylic Acid 1,4-Dioxides as Antimycobacterial Agents: Mechanistic Studies and Therapeutic Potential

**DOI:** 10.3390/ph16111565

**Published:** 2023-11-06

**Authors:** Svetlana G. Frolova, Aleksey A. Vatlin, Dmitry A. Maslov, Buhari Yusuf, Galina I. Buravchenko, Olga B. Bekker, Ksenia M. Klimina, Svetlana V. Smirnova, Lidia M. Shnakhova, Irina K. Malyants, Arseniy I. Lashkin, Xirong Tian, Md Shah Alam, George V. Zatonsky, Tianyu Zhang, Andrey E. Shchekotikhin, Valery N. Danilenko

**Affiliations:** 1Laboratory of Bacterial Genetics, Vavilov Institute of General Genetics, Russian Academy of Sciences, 119333 Moscow, Russia; vatlin_alexey123@mail.ru (A.A.V.); obbekker@mail.ru (O.B.B.); ppp843@yandex.ru (K.M.K.); vsmirnov64@mail.ru (S.V.S.); valerid@vigg.ru (V.N.D.); 2Phystech School of Biological and Medical Physics, Moscow Institute of Physics and Technology (State University), 141701 Dolgoprudny, Russia; 3Institute for Regenerative Medicine, Department of Dermatology and Venereology, Sechenov First Moscow State Medical University (Sechenov University), 119991 Moscow, Russia; shnakhova_l_m@staff.sechenov.ru; 4Institute of Ecology, Peoples’ Friendship University of Russia (RUDN University), 117198 Moscow, Russia; 5Division of Gastroenterology and Hepatology, Department of Medicine, Stanford University School of Medicine, Stanford, CA 94305, USA; dmaslov@stanford.edu; 6State Key Laboratory of Respiratory Disease, Guangzhou Institutes of Biomedicine and Health, Chinese Academy of Sciences, Guangzhou 510530, China; yusuf@gibh.ac.cn (B.Y.); tian_xirong@gibh.ac.cn (X.T.); alam@gibh.ac.cn (M.S.A.); 7University of Chinese Academy of Sciences, Beijing 100049, China; 8Guangdong-Hong Kong-Macao Joint Laboratory of Respiratory Infectious Diseases, Guangzhou Institutes of Biomedicine and Health, Chinese Academy of Sciences, Guangzhou 510530, China; 9China-New Zealand Joint Laboratory on Biomedicine and Health, Guangzhou Institutes of Biomedicine and Health, Chinese Academy of Sciences, Guangzhou 510530, China; 10Gause Institute of New Antibiotics, 119021 Moscow, Russia; buravchenkogi@gmail.com (G.I.B.); gzatonsk@gmail.com (G.V.Z.); shchekotikhin@mail.ru (A.E.S.); 11Lopukhin Federal Research and Clinical Center of Physical-Chemical Medicine of Federal Medical Biological Agency (Lopukhin FRCC PCM), 119435 Moscow, Russia; iricam@mail.ru; 12Federal Research and Clinical Center of Physical-Сhemical Medicine, 119435 Moscow, Russia; arseniy.lashkin@gmail.com; 13Laboratory of Molecular Oncology, Department of Bioorganic Chemistry, Faculty of Biology, Lomonosov Moscow State University, 119991 Moscow, Russia

**Keywords:** derivatives of quinoxaline-2-carboxylic acid 1,4-dioxide, antimicrobial activity, drug resistance, *M. tuberculosis*, *M. smegmatis* mutants

## Abstract

The World Health Organization (WHO) reports that tuberculosis (TB) is one of the top 10 leading causes of global mortality. The increasing incidence of multidrug-resistant TB highlights the urgent need for an intensified quest to discover innovative anti-TB medications In this study, we investigated four new derivatives from the quinoxaline-2-carboxylic acid 1,4-dioxide class. New 3-methylquinoxaline 1,4-dioxides with a variation in substituents at positions 2 and 6(7) were synthesized via nucleophilic aromatic substitution with amines and assessed against a *Mycobacteria* spp. Compound **4** showed high antimycobacterial activity (1.25 μg/mL against *M. tuberculosis*) and low toxicity in vivo in mice. Selection and whole-genomic sequencing of spontaneous drug-resistant *M. smegmatis* mutants revealed a high number of single-nucleotide polymorphisms, confirming the predicted mode of action of the quinoxaline-2-carboxylic acid 1,4-dioxide **4** as a DNA-damaging agent. Subsequent reverse genetics methods confirmed that mutations in the genes MSMEG_4646, MSMEG_5122, and MSMEG_1380 mediate resistance to these compounds. Overall, the derivatives of quinoxaline-2-carboxylic acid 1,4-dioxide present a promising scaffold for the development of innovative antimycobacterial drugs.

## 1. Introduction

Tuberculosis (TB), an infectious disease caused by the pathogen *Mycobacterium tuberculosis*, remains a significant global health challenge. When patients adhere to a correctly selected treatment regimen, the cure rates hover around 85% [1]. However, factors such as inappropriate chemotherapy, patient non-compliance, HIV infection, and diabetes contribute to the development of TB drug resistance [2]. Clinical cases of *M. tuberculosis* resistant to the latest approved antituberculosis drugs, bedaquiline and pretomanid, have already been reported [3]. These drug-resistant TB strains are challenging to treat, necessitating lengthier treatment regimens that often come with more severe side effects. This underscores the urgent need to search for and develop new antituberculosis drugs that employ novel mechanisms of action [4].

It is well known that the presence of two *N*-oxide fragments in the structure of quinoxaline 1,4-dioxides (QdNOs) endows them a wide spectrum of biological properties, including antitumor, antibacterial, antiparasitic, anti-inflammatory, antioxidant, and herbicidal activities [5]. Moreover, QdNOs exhibit high inhibitory activity against various pathogenic microorganisms such as *Escherichia coli*, *Pasteurella multocida*, dysentery spirochete [6], and *Brachyspira hyodysenteriae* [7]. Several studies have confirmed the mechanism of antibacterial action of QdNOs associated with the production of free radicals during bio-reduction that lead to DNA damage [8]. Furthermore, some derivatives of QdNOs have also exhibited excellent inhibitory activity against *M. tuberculosis*, indicating the great interest in this scaffold for the development of new anti-TB drugs [9].

Recent in silico studies of quinoxaline 1,4-dioxides binding to mycobacterial DNA gyrase have indicated that these compounds can bind to the same site as the well-established DNA–gyrase ligand, novobiocin [10]. Some derivatives of this class are currently undergoing preclinical trials, including studies on MDR *M. tuberculosis* strains [11].

In prior research, we described a series of novel 2-acyl-3-trifluoromethylquinoxaline 1,4-dioxides with significant inhibitory properties against *M. smegmatis*. However, their activity against *M. tuberculosis* was limited [12]. In this study, we introduce a novel series of quinoxaline-2-carboxylic acid 1,4-dioxide derivatives with substituent variations at positions 3, 6, and 7. These compounds underwent testing against *M. smegmatis* and the autoluminescent *M. tuberculosis* H37Ra (AlRa) strains. Compound **4**, or 7-chloro-2-(ethoxycarbonyl)-3-methyl-6-(piperazin-1-yl)quinoxaline 1,4-dioxide, exhibited the most potent activity against both *M. tuberculosis* and *M. smegmatis* strains, leading to its selection as the lead compound for further studies on its mechanism of action and resistance. A comparative genomic analysis of spontaneous *M. smegmatis* mutants resistant to compound 4 pinpointed mutations in genes that encode the redox enzymes of pyruvate metabolism, which might function as pro-drug activators. Using reverse genetics, we verified that the genes *MSMEG_5122* and *MSMEG_4646* confer resistance to quinoxaline in *M. smegmatis*. Additionally, resistance to compound **4** can also be attributed to the MmpS5-MmpL5 efflux. In vivo tests of the compounds at specified doses in a mouse infection model yielded results comparable to the negative control. However, no toxic effects were detected. Consequently, quinoxaline 1,4-dioxide **4** has emerged as a promising foundation for refining anti-TB drug candidates, emphasizing the enhancement of its in vivo activity.

## 2. Results

### 2.1. Chemistry

The synthesis of the designed QdNOs **4**–**7**, **9** are presented in Figure 1 and Figure 2. The synthetic route included the Beirut reaction between 5,6-dichlorobenzofuroxan (**1**) with 1,3-dicarbonyl compounds in the presence of K_2_CO_3_ as the catalyst and further treatment of the formed 6,7-dichloro-3-methylquinoxaline 1,4-dioxides **2**, **3** with cyclic amines in DMF at 50 °C that resulted in target 6-aminosubstituted derivatives **4**–**6** (Figure 1). The revealed regioselectivity of the nucleophilic substitution of the chlorine atom at position 6 of quinoxalines **2** and **3** (Figure 1) can be explained by the conjugation of the C6 carbon atom with the electron-withdrawing group at position 2 of the heterocycle of these derivatives. This conjugation leads to an increase in the positive charge at C6 carbon atom as well as stabilization of the intermediate Meisenheimer complex, thereby enhancing its reactivity in nucleophilic substitution reactions [13]. The structure of all synthesized compounds **4**–**7** (Figure 1) was characterized by using HRMS ESI, HPLC, and NMR spectroscopies. 

It was found that the chlorine atom at the 7-position of quinoxaline 2 also undergoes substitution with an amine under these conditions. So, the heating of dichloroderivative **2** with an excess of piperazine in DMF leads to the formation of the byproduct—7-amino-substituted 2-carboethoxy-3-methylquinoxaline 1,4-dioxide **7** at a yield of 21% (Figure 1). 

At the final stage, free amines **4**, **5**, and **7** were transformed into target hydrochlorides by treatment of a solution of hydrochloric acid in methanol. The purity (>95%) of the target compounds was determined by HPLC. The structures of isomeric quinoxalines **4** and **7** were unambiguously confirmed by an analysis of the increment values of the signals in the ^13^C NMR spectra, as described earlier for quinoxaline-2-carbonitriles [13,14]. Using two-dimensional NMR spectroscopy (HSQC and HMBC), the ^1^H and ^13^C NMR signals of the regioisomeric derivatives **4** and **7** were assigned (Appendix A). Comparison of the ^13^C NMR spectra of isomers **4** and **7** (Appendix A, respectively) with that of the unsubstituted 2-ethoxycarbonyl-3-methylquinoxaline 1,4-dioxide (Appendix A) revealed negative shifts in the signals of the carbon atoms at positions 9 and 10 for the 6-aminoderivative **4** and 7-aminoderivative **7**, respectively. These shifts are consistent with previously obtained results. These shifts also correlated with the increments of substituents in these positions, characteristic of benzene derivatives and polyaromatic compounds (Appendix A) [13].

The positions of the substituents have been definitely confirmed via 2D NMR using HMBC and NOESY experiments. A key aspect of the structure elucidation was the selective NOESY experiment using the signal of the CH_3_ group at position 3 of both isomers for selective excitation. In the case of the 6-isomer (compound **4**), an enhancement of the high-field signal (at 7.89 ppm) corresponding to the CH-group in the *ortho*-position to the piperazine substituent, was observed (Appendix A). Conversely, the same selective NOESY experiment revealed a lower field signal enhancement (at 8.41 ppm) for the 7-isomer (compound **7**) (Appendix A). These observations provided evidence that the piperazine substituent is located at position 6 of the quinoxaline ring in derivative **4** and at position 7 in its isomer **7**, respectively. Additionally, crucial correlations in HMBC spectra of regioisomers were observed: a four-bond coupling between signals H8 and C2 for the 7-isomer and between H5 and C3 for the 6-isomer. This unequivocally verified the proposed structures of the regioisomers (Appendix A, respectively).

It is worth noting that compounds **4**–**7** exhibited remarkable stability in the solid state and were quite stable in aqueous solutions. A store of aqueous solutions of derivatives **4**, **6**, and **7** at 37 °C for 7 days revealed a slow degradation of these compounds that did not exceed 10% (Appendix A). Surprisingly, a significantly higher decay rate was observed in these conditions for carboxamide **5**.

A previously described 3-trifluoromethylquinoxaline 1,4-dioxide derivative **9** [12] was prepared in the same way, which involved two steps: a cyclization of benzofuroxan with ethyl 4,4,4-trifluoroacetoacetate in chloroform in the presence of triethylamine and a subsequent substitution of chlorine atom in position 7 of 3-trifluorimethylquinoxaline 1,4-dioxide **8** with *N*-Boc-piperazine in THF (Figure 2). The ^13^C NMR spectra of derivatives **8** and **9** had two characteristic quadruplet signals that corresponded to their structures. These signals had chemical shifts at δ = 127–129 ppm (*J* = 33.7 Hz) and δ = 118.8–118.9 ppm (*J* = 272–274 Hz), corresponding to the carbon atom in position 3 of the heterocycle and the associated trifluoromethyl group, respectively (compounds **8**–**9**, Figure 2; Appendix A).

The deprotection of Boc-intermediate **9** with HCl in methanol led to the hydrochloride of 6-chloro-2-ethoxycarbonyl-3-trifluoromethyl-7-(piperazin-1-yl)quinoxaline 1,4-dioxide (**9**, Figure 2).

### 2.2. Biology 

#### 2.2.1. Biological Screening of Quinoxaline 1,4-Dioxides Derivates on *M. smegmatis* and *M. tuberculosis*

Antibacterial activity of the novel compounds was tested against Gram-positive bacteria strains, specifically *M. smegmatis* mc^2^ 155 and autoluminescent *M. tuberculosis* H37Ra. The minimum inhibitory concentration (MIC) values of compounds **4**–**9** and two reference drugs are summarized in Table 1. 

A comparative assessment of the screening outcomes highlighted the pivotal role of substituents in determining the antimicrobial properties of the quinoxaline 1,4-dioxides **4**–**7**. Specifically, compound **4**, which contains an ethoxycarbonyl group at position 2 of quinoxaline, displayed the most pronounced activity across all tested strains. Substituting the ethoxycarbonyl group at position 2 with a carboxamide group of analogous electronic properties negatively influenced the antitubercular effectiveness of the compounds (derivatives **5** and **6**) in both test cultures. Intriguingly, the inclusion of a lipophilic piperidine segment at position 6 (in compound **6**) marginally enhanced the antimycobacterial activity against the *M. tuberculosis* AlRa strain compared to its counterpart, compound **5**. The location of the amino group within the benzene ring of the quinoxaline core significantly influenced the antimicrobial activity. For instance, moving the piperazine fragment from position 6 to 7 led to a drop in the antitubercular activity of derivative **7** when compared to its regioisomer **4**. Furthermore, the lead compound **4** exhibited activity levels against both mycobacterial strains that were comparable to the reference drug rifampicin (MIC values of 0.03 and 4 µg/mL, respectively). Additionally, derivatives **4**–**7** showed activity levels akin to the reference drug dioxidine (MIC values ranging between 10–16 µg/mL). As a result, compound **4** was selected for the subsequent identification of spontaneous resistant mutants. 

#### 2.2.2. *M. smegmatis* Drug-Resistant Mutants and Their Genomic Analysis

The fast-growing and non-pathogenic bacterial strain *M. smegmatis* mc^2^ 155 serves as a more suitable model for obtaining spontaneous resistant mutants, studying bacterial resistance, and understanding the drug’s mechanism of action. We managed to procure spontaneous mutants of *M. smegmatis* mc^2^ 155 resistant to quinoxaline 1,4-dioxide **4** at a frequency of 2.3 × 10^−8^. While we made several attempts, we were unsuccessful in obtaining genuinely resistant *M. tuberculosis* mutants. Although individual colonies appeared on plates with higher concentrations of compound **4**, their MIC values mirrored that of the parent strain.

For in-depth analysis, we randomly selected five mutants to study their cross-resistance and to perform whole-genome sequencing. Comparative genomic analysis highlighted that the mutants qdr1, qdr2, and qdr5 had between 31 and 35 unique, non-synonymous mutations, while qdr3 and qdr4 had 27 and 18, respectively (see Appendix B, Appendix A).

Common mutations were identified in four genes: *MSMEG_1380*, *MSMEG_4646*, *MSMEG_4648*, and *MSMEG_5122*. The qdr1 mutant strain displayed an SNP in *MSMEG_4648* and an insertion of two nucleotides in the promoter region of *MSMEG_5122*. Meanwhile, the qdr2 mutant showcased an SNP directly in *MSMEG_5122*. The *MSMEG_5122* and *MSMEG_4648* genes encode ferredoxin and a DNA-binding protein, respectively. The remaining mutants (qdr3-5) exhibited diverse single SNPs in *MSMEG_4646*, which encodes pyruvate synthase, playing a role in glycolysis/gluconeogenesis processes (Table 2).

Additionally, all the derived strains, barring qdr4, exhibited mutations in *MSMEG_1380*. This gene encodes the AcrR/TetR_N transcriptional repressor and influences the expression of the MmpS5-MmpL5 efflux system. It is noteworthy that our prior observations showed mutations in *MSMEG_1380* enhancing the expression of the MmpS5-MmpL5 efflux system [15]. This system mediates resistance across various mycobacterial species to a wide spectrum of antibiotics, such as bedaquiline and clofazimine [16], azoles [17], thiacetazone derivatives [18], imidazo [1,2-b][1,2,4,5]tetrazines [15], and tryptanthrins [19].

#### 2.2.3. Investigating the Effect of Individual Gene Mutations on Sensitivity to Quinoxalin-1,4-dioxide

To understand the phenotype of the spontaneous resistant mutants, we assessed the MICs of synthesized compounds on the resistant mutants *M. smegmatis* qdr1-qdr5. All displayed an increased MIC for quinoxaline 1,4-dioxide **4**, ranging from 2–4 times. The qdr1, qdr2, and qdr3 strains exhibited resistance to its analogues **5**–**7** and **DIOX**. Rifampicin was used as a control. 

Utilizing the method of homologous recombination via the suicide vector system p2Nil/pGOAL19 [20], we constructed *M. smegmatis* strains with the identified mutations. This led to three recombinant strains:*M. smegmatis* 4646c (carrying mutation AAC95CAC in *MSMEG_4646*, analogous to *M. smegmatis* qdr4);*M. smegmatis* 4648c (with mutation CAG49CCG in *MSMEG_4648*, matching *M. smegmatis* qdR1);*M. smegmatis* 5122c (harboring mutation AT(-72-71)GA in the *MSMEG_5122* promoter, akin to *M. smegmatis* qdR1).

The recombinant strains *M. smegmatis* 4646c and *M. smegmatis* 5122c showed more than double the MIC values of compound **4** and **DIOX**. Both also displayed resistance to compound **7**, while *M. smegmatis* 5122c had heightened MICs for compounds **5** and **6**. In contrast, the MSMEG_4648 mutation did not affect sensitivity to compound **4** but did enhance resistance to **DIOX**. The MIC values for these compounds and references are detailed in Table 3.

Given the identified mutation in *MSMEG_1380*—which encodes the transcriptional regulator of the *mmpS5-mmpL5* operon—we assessed the susceptibility of our synthesized compounds to MmpS5-MmpL5 efflux. 

Utilizing our previously developed test system based on three *M. smegmatis* strains with varying MmpS5-MmpL5 expression [21], we observed *M. smegmatis* mc^2^ 155 and *M. smegmatis* atr9c showing identical growth inhibition halos. However, *M. smegmatis* Δmmp5 was more susceptible to compounds **4**–**6**. It appears that the MmpS5-MmpL5 efflux offers basal resistance to compound **4** and some derivatives, but its overexpression (as seen in *M. smegmatis* atr9c) does not heighten this resistance. Compound **7** seems unaffected by this efflux, likely due to substituent positions influencing efflux affinity. The growth inhibition halo diameters are depicted in Figure 2.

The capability of our synthesized amino derivatives of quinoxaline-2-carboxylic acid 1,4-dioxide to bypass the transmembrane transporters was studied using *M. smegmatis* Δmmp5 and *M. smegmatis* atr9c. These compounds exhibited promising activity against the multidrug-resistant *M. smegmatis* atr9c strain, known for overexpressing *mmpS5-mmpL5* genes. Furthermore, compounds **4**–**7** also displayed comparable activity against both the wild-type *M. smegmatis* mc^2^ 155 and its resistant counterpart *M. smegmatis* Δmmp5 (see Figure 2). Intriguingly, the amino group’s position in the quinoxaline core plays a crucial role in drug penetration, with regioisomer **7** of compound **4** showcasing similar activity levels in both *M. smegmatis* Δmmp5 and the original *M. smegmatis* mc^2^ 155 test cultures.

#### 2.2.4. Mutations in Genes MSMEG_5122 and MSMEG_4648 Change Their Expression

The mutation in *MSMEG_4648* led to a noticeable increase in the expression of genes *MSMEG_4648* and *MSMEG_4645* (by 38% and 52%, respectively) (*p* < 0.01, Figure 3). However, the expression of *MSMEG_4646* remained unchanged. This change in expression might not be sufficient to confer full resistance, as seen in the drug sensitivity profile of the *M. smegmatis* 4648c strain. Conversely, in the *M. smegmatis* 5122c strain, a promoter mutation caused the expression of gene *MSMEG_5122* to drop significantly (by 2.76 times) (*p* < 0.01, Figure 2), resulting in resistance to quinoxaline 1,4-dioxides.

#### 2.2.5. Derivative 4 Displays Minimal Cytotoxicity on Human Fibroblasts

In primary cultures of human fibroblasts, the cytotoxicity of derivative **4** was assessed using MTT analysis and IC_50_ determinations. The IC_50_ value for **4** was 67 μM. For comparison, **DIOX** and mitoxantrone were used, yielding IC_50_ values of 3000 μM and 0.5 μM, respectively.

An essential metric is the selectivity index (SI = IC_50_/MIC). A compound is deemed promising if its SI is above 10. Derivative **4** displayed an SI of 53.6, whereas **DIOX**’s SI stood at 300. While derivative **4** had a lower SI than **DIOX**, its value was still above 10, underscoring the potential of further investigations into its antituberculosis properties [22].

#### 2.2.6. In Vivo Study Activity of Derivative 4 in Murine Tuberculosis Model

Mice infected with the autoluminescent reporter strain of *M. tuberculosis* AlRa were treated with derivative **4** both orally and intraperitoneally (I.P.). Rifampicin (10 mg/kg) served as a positive control. The positive control exhibited slower activity but remained significantly different from the test drug across all tested doses (Figure 4).

While derivative **4** did not exhibit immediate efficacy at any dosage, no toxicity was observed, as all treated mice survived. A statistical analysis using Dunnett’s multiple test indicated that the drug’s activity could be influenced by the administration route (Appendix A).

## 3. Discussion

Quinoxaline is known for its biological and pharmacological activity [5]. In this study, we detailed the synthesis and antimicrobial potency of new quinoxaline 1,4-dioxide derivatives.

We evaluated the antimicrobial activity of these compounds in vitro using *M. smegmatis* and *M. tuberculosis* AlRa cultures. Derivative **4** stood out, demonstrating the lowest MICs for both cultures: 1.25 and 4 μg/mL, respectively. Consequently, we chose derivative **4** for further cytotoxicity and in vivo analyses. The MTT assay on human fibroblasts showed derivative **4**’s SI to be 53.6. Notably, any compound with an SI ≥ 10 is regarded as a promising candidate for further exploration.

Our in vivo study on mice infected with the autoluminescent *M. tuberculosis* AIRa revealed no early activity for derivative **4** at any dosage. Still, there were no recorded mortalities. Interestingly, intraperitoneal administration appeared slightly more effective than oral administration, suggesting potential challenges with oral assimilation for derivative **4**.

Detailed study of the mode of actions of QdNOs will provide information about their biological targets and the mechanism of action and will be valuable to construct models to screen new drugs. Whole-genome analysis revealed numerous mutations in all obtained drug-resistant strains, which indicated the high mutagenic potential of derivative **4**. One of the possible mechanisms of action of quinoxaline 1,4-dioxide derivatives is the direct introduction of single- and double-stranded breaks in the DNA of bacteria, which could explain the numerous mutations in *M. smegmatis* qdr strains [23]. Despite this, the exact mechanism of action of QdNOs has not yet been established. We identified four mutations in genes *MSMEG_1380*, *MSMEG_4646*, *MSMEG_4648*, and *MSMEG_5122* that were presented in several qdr-mutants simultaneously. We have previously shown that mutations in *MSMEG_1380*, which encodes a transcriptional regulator, lead to the overexpression of MmpS5-MmpL5. This overexpression can result in mycobacterial resistance to a wide range of antibiotics, including thiacetazone, bedaquiline and clofazimine [16], imidazo [1,2-b][1,2,4,5]tetrazines [15], and tryptanthrin [19]. To confirm the role of MmpS5-MmpL5 in drug resistance, all synthesized compounds were screened in our previously described test system [21]. *M. smegmatis* Δmmp5 was more sensitive to compounds **4**–**6,** while *M. smegmatis* mc^2^ 155 and *M. smegmatis* atr9c had the same level of resistance to all compounds. The fact that overexpression of MmpS5-MmpL5 in the atr9c strain did not result in increased resistance but absence of efflux resulted in decreased resistance might indicate that MmpS5-MmpL5 can provide a basal level of resistance. 

Through a reverse genetics approach, we confirmed that mutations in *MSMEG_4646* and in the promoter region of *MSMEG_5122* mediated resistance of *M. smegamtis* to the described QdNOs. *MSMEG_4646* encodes the alpha subunit of ferredoxin oxidoreductase (pyruvate synthase) involved in pyruvate metabolism. *MSMEG_5122* encodes ferredoxin, which acts as an electron acceptor for pyruvate synthase during the process of oxidation of pyruvate to acetyl-CoA. *MSMEG_4648* is annotated as a DNA-binding protein and can be a transcriptional regulator of the nearby operon, which encodes the alpha and beta subunits of the mentioned pyruvate synthase. The role of *MSMEG_4648* was not obvious because the mutation in this gene was observed with a mutation in the promoter region of *MSMEG_5122* simultaneously. The measured MIC values of the constructed strain *M. smegmatis* 4646c were the same as that of the w.t. strain to all characterized compounds. 

We established that mutations in the promoter of *MSMEG_5122* and in *MSMEG_4648* changed the expression of *MSMEG_5122* and *MSMEG_4645* with *MSMEG_4648*, respectively. However, the expressions of *MSMEG_4645* and *MSMEG_4648* were altered insignificantly, while *MSMEG_5122* was decreased by almost 3 times. Thus, the reduced expression of ferredoxin encoded by *MSMEG_5122* resulted in increased resistance, which might indicate that ferredoxin takes part in the activation of **4** or in the metabolic pathways of its mechanism of action. Meanwhile, pyruvate synthase encoded by *MSMEG_4646* can participate in the inactivation of derivative **4**.

In conclusion, QdNOs demonstrate potential as valuable scaffolds for new drug development. The action mechanism and biological targets of QdNOs remain subjects of debate. However, our findings suggest that these compounds have significant bacterial DNA-damage capabilities and impact pyruvate oxidation metabolic pathways. Further research involving the transcriptomic analysis of *M. smegmatis* mc^2^ 155 treated with derivative **4** at various concentrations and time spans may provide additional insights on QdNOs’ mechanism of action. 

## 4. Materials and Methods

### 4.1. Materials and General Methods

NMR spectra were recorded on a Varian Mercury 400 Plus instrument operated at 400 MHz (^1^H NMR) and 100 MHz (^13^C NMR) or a Bruker AVANCE III 500 (Bruker Biospin, Rheinstetten, Germany) NMR spectrometer equipped with a broadband Z-gradient probehead with a direct observe BB coil (PABBO) at 500.18 MHz for ^1^H and 125.77 MHz for ^13^C, respectively. Chemical shifts were measured in DMSO-*d_6_* using TMS as an internal standard. Spectra for all obtained compound were recorded in DMSO-*d_6_* solutions at 303 K and were referenced against residual solvent signals: 2.50 ppm for DMSO-d5 for 1H and 39.50 ppm for DMSO-*d_6_* for ^13^C, respectively. The 1D and 2D NMR spectra were processed using TopSpin 3.2 Bruker (Bruker Biospin, Rheinstetten, Germany) or ACD Laboratories Spectral Processor Academic Edition (Advanced Chemistry Development. ACD, Inc., Toronto, ON, Canada, 2012. Software program). The ^1^H and ^13^C signal assignment was conducted by using 1H{^13^C} HSQC, 1H{^13^C} HMBC, and 2D NOESY NMR experimental data. Standard pulse sequences were used. For 2D NOESY and selective NOESY experiments, mixing times of 400 and 600 ms were correspondingly used. For selective excitation, an 80 ms Gaussian-shaped pulse was used. Analytical TLC was performed on silica gel F254 plates (Merck, Rahway, NJ, USA) and column chromatography on Silica Gel Merck 60. Melting points were determined on a Buchi SMP-20 apparatus and are uncorrected. High-resolution mass spectra were recorded via electron spray ionization on a Bruker Daltonics micro OTOF-QII instrument. UV spectra were recorded on a Hitachi-U2000 spectrophotometer (Hitachi High-Technologies Corporation (HHC), Hitachinaka, Japan). IR spectra were recorded on Nicolet iS10 Fourier transform IR spectrometer (Nicolet iS10 FT-IR, Thermo Scientific, Madison, WI, USA). The purity of compounds **4**–**7**, **9** was >95% as determined by HPLC analysis. All products were dried under vacuum at room temperature. HPLC was performed using a Shimadzu Class-VP V6.12SP1 system (GraseSmart RP-18, 6 × 250 mm) with eluents as follows: A, H_3_PO_4_ (0.01 M); B, MeCN. All solutions were evaporated at reduced pressure on a Buchi-R200 rotary evaporator (Büchi Labortechnik AG, Postfach, Switzerland) at a temperature below 50 °C. All products were dried under vacuum at room temperature. All solvents, chemicals, and reagents were obtained from Sigma-Aldrich (St. Louis, MO, USA) (unless specified otherwise) and used without purification. 

### 4.2. Synthesis

#### 4.2.1. General Procedure for Synthesis of Compounds **2**–**9**

The quinoxalines **2**, **3** and **8**, **9** used in this study were previously described and synthesized in [24]. The preparation of the compounds **2**, **3** and **8** was carried out by the Beirut reaction as presented in Figure 1. The starting 5,6-dichlorobenzofuroxan (**1**) was obtained by using a previously described method [25]. 

#### 4.2.2. 2-(Ethoxycarbonyl)-3-methylquinoxaline 1,4-dioxide

This compound was prepared from unsubstituted benzofuroxan and acetoacetic ester as described for **3**. It was a yellow powder (yield 23%, mp 105–106 °C). ^1^H NMR (400 MHz, DMSO-*d_6_*) δ 8.40–8.35 (2H, m, H-8, H-5); 7.97–7.90 (2H, m, H-7, H-6); 4.50 (2H, q, *J* = 7.2, O**CH_2_**CH_3_); 2.41 (3H, s, CH_3_); 1.36 (3H, t, *J* = 7.0, OCH_2_**CH_3_**). ^13^C NMR (100 MHz, DMSO-*d_6_*) δ159.7(CO); 138.3 (3-C); 137.4 (9-C); 136.2 (10-C); 134.8 (2-C); 132.7 (6-C); 131.6 (7-C); 119.6 (5-CH); 119.5 (8-CH); 63.2 (O**CH_2_**CH_3_); 14.1 (CH_3_); 13.8 (OCH_2_**CH_3_**).

#### 4.2.3. 6,7-Dichloro-2-(ethoxycarbonyl)-3-methylquinoxaline 1,4-dioxide (**2**)

The mixture of acetoacetic ester (0.5 mL, 3.6 mmol) and K_2_CO_3_ (0.14 g, 1 mmol) in acetone (5 mL) was added to a solution of the 5,6-dichlorobenzofuroxan (0.5 g, 2.4 mmol) in acetone (5 mL), and the reaction mixture was stirred at room temperature overnight. After evaporation, the solution (a crude solid) was precipitated from a dichloromethane-hexane mixture (3:1) and washed with diethyl ether. The obtained yellow precipitate was purified via recrystallization from methanol. The yield of the target compound was 70% (mp 222–223 °C). ^1^H NMR (400 MHz, DMSO-*d_6_*) δ 8.61 (1H, s, H-8); 8.58 (1H, s, H-5); 4.49 (2H, q, *J* = 7.0, O**CH_2_**CH_3_); 2.40 (3H, s, CH_3_); 1.33 (3H, t, *J* = 7.0, OCH_2_**CH_3_**). ^13^C NMR (100 MHz, DMSO-*d_6_*) δ159.5(CO); 139.9 (3-C); 137.1 (9-C); 136.4 (10-C); 135.9 (7-CCl); 135.8 (6-CCl); 135.4 (2-C); 121.63 (8-CH); 121.56 (5-CH); 63.7 (O**CH_2_**CH_3_); 14.4 (CH_3_); 14.0 (OCH_2_**CH_3_**). 

#### 4.2.4. 2-Carbamoyl-6,7-dichloro-3-methylquinoxaline 1,4-dioxide (**3**)

A mixture of acetoacetamide (0.2 g, 2.2 mmol) and K_2_CO_3_ (80 mg, 0.6 mmol) in ethanol (3 mL) was added to a solution of 5,6-dichlorobenzofuroxan (0.3 g, 1.4 mmol) in ethanol (3 mL), and the reaction mixture was stirred at room temperature overnight. Then, the reaction mixture was cooled, and the product filtered and recrystallized from ethanol to give compound **3** (0.2 g, 58%, yellow solid, mp > 260 °C). ^1^H NMR (400 MHz, DMSO-*d_6_*) δ 8.65 (1H, s, H-8); 8.64–8.61 (2H, br. m, CONH_2_); 8.26 (1H, s, H-5); 2.43 (3H, s, CH_3_). ^13^C NMR (100 MHz, DMSO-*d_6_*) δ160.6(CO)140.6 (3-C); 139.5 (2-C); 136.5 (9-C); 136.3 (10-C); 136.0 (7-CCl); 135.6 (6-CCl); 121.9 (8-CH); 121.7 (5-CH); 14.9 (CH_3_).

#### 4.2.5. 7-Chloro-2-(ethoxycarbonyl)-3-methyl-6-(piperazin-1-yl)quinoxaline 1,4-dioxide hydrochloride (**4**)

Piperazine (407 mg, 4.7 mmol) was added to a solution of 6,7-dichloro-2-ethoxycarbonyl-3-methylquinoxaline 1,4-dioxide (**2**, 300 mg, 0.95 mmol) in DMF (15 mL), and the mixture was stirred at 50 °C for 10 h. After the reaction was completed (as determined by TLC), the reaction mixture was poured into cold water (30 mL) and cooled. The precipitate was filtered and washed with water (30 mL). The filtrate from the reaction mixture after crystallizations of the product **4** were collected and evaporated under reduced pressure. The crude mixture of regioisomers **4** and 7 was separated via column chromatography (chloroform—ethyl acetate, 5:1). The collected crude product **4** was crystallized from ethanol, which gave pure 6-isomer **4** (0.23 g, 66%). The residue obtained after crystallization was dissolved in a mixture of water (3 mL) and hydrochloric acid (0.4 mL, 6.3 mmol). The hot solution was filtered, and the product was precipitated with a methanol–acetone–diethyl ether mixture (1:2:5). The yellow solid was collected via filtration; washed with acetone, diethyl ether, and n-hexane successively; and dried. The yield of the hydrochloride of **4** was 220 mg (63%) as an orange powder (mp 254–255 °C (dec.)). HPLC (LW = 377 nm, gradient B 10/50% (45 min)) t_R_ = 14.5 min, purity 94.2%. λ_max._, EtOH: 217, 277, 326, 377 nm. IR ν_max_, (film) cm^−1^ 3479, 3030, 2840, 1733, 1599, 1519, 1476, 1451, 1389, 1350, 1318, 1262, 1235, 1188, 1148, 1085, 1059, 1040, 938, 883, 859, 774. ^1^H NMR (500 MHz, DMSO-*d_6_*) δ 9.75 (2H, br. s, NH_2_^+^); 8.37 (1H, s, H-8); 7.89 (1H, s, H-5); 4.49 (2H, q, *J* = 7.0, O**CH_2_**CH_3_); 3.47–3.44 (4H, br. m, 2 × CH_2_); 3.30–3.28 (4H, br. m, 2 × CH_2_); 2.40 (3H, s, CH_3_); 1.35 (3H, t, *J* = 7.0, OCH_2_**CH_3_**). ^13^C NMR (125 MHz, DMSO-*d_6_*) δ159.5(CO); 151.5 (6-C); 139.0 (3-C); 137.0 (10-C); 134.1 (2-C); 132.6 (7-CCl); 132.3 (9-C); 121.2 (8-CH); 108.8 (5-CH); 63.3 (O**CH_2_**CH_3_); 47.7 (2 × CH_2_); 42.9 (2 × CH_2_); 14.1 (CH_3_); 13.7 (OCH_2_**CH_3_**). HRMS (ESI) calculated for C_16_H_20_ClN_4_O_4_^+^ [M+H]^+^ 367.1168, found 367.1156.

#### 4.2.6. 2-Carbamoyl-7-chloro-3-methyl-6-(piperazin-1-yl)quinoxaline 1,4-dioxide hydrochloride (**5**)

This compound was prepared from **3** and piperazine as described for **4**. It was a yellow powder (yield 69%, mp 252–254 °C (dec.)). HPLC (LW = 377 nm, gradient B 7/40% (45 min)) t_R_ = 3.6 min, purity 95.0%. λ_max._, EtOH: 229, 277, 316, 374 nm. IR ν_max_, (film) cm^−1^ 3381, 2455, 1681, 1596, 1566, 1516, 1446, 1378, 1341, 1302, 1257, 1235, 1152, 1133, 1084, 1058, 1027, 961, 919, 864, 829, 756. ^1^H NMR (400 MHz, DMSO-*d_6_*) δ 9.66 (2H, br. s, NH_2_^+^); 8.36 (2H, d, *J* = 13.6, CONH_2_); 8.30 (1H, s, H-8); 7.88 (1H, s, H-5); 3.55–3.43 (4H, br. m, 2 × CH_2_); 3.33–3.25 (4H, br. m, 2 × CH_2_); 2.43 (3H, s, CH_3_). ^13^C NMR (100 MHz, DMSO-*d_6_*) δ160.7(CO); 150.8 (6-C); 139.7 (3-C); 137.9 (9-C); 136.5 (10-C); 132.8 (7-CCl); 132.4 (2-C); 121.5 (8-CH); 109.0 (5-CH); 47.8 (2 × CH_2_); 42.9 (2 × CH_2_); 13.9 (CH_3_). HRMS (ESI) calculated for C_14_H_17_ClN_5_O_3_^+^ [M+H]^+^ 338.1014, found 338.1001.

#### 4.2.7. 2-Carbamoyl-7-chloro-3-methyl-6-(piperidin-1-yl)quinoxaline 1,4-dioxide (**6**)

Piperidine (340 mL, 3.5 mmol) was added to a solution of 2-carbamoyl-6,7-dichloro-3-methylquinoxaline 1,4-dioxide (**3**, 0.2 g, 0.69 mmol) in DMF (15 mL), and the mixture was stirred at 50 °C for 10 h. After the reaction was completed (as determined by TLC), the reaction mixture was poured into cold water (30 mL) and cooled. The precipitate was filtered and washed with water (30 mL). The crude product was purified via flash chromatography on a silica gel using eluting solvent (toluene–diethylether mixture, 6:1). It was an orange powder (yield 47%, mp 154–155 °C). HPLC (LW = 274 nm, gradient B 30/50% (45 min)) t_R_ = 12.9 min, purity 99.7%. λ_max._, EtOH: 220, 279, 333, 377 nm. IR ν_max_, (film) cm^−1^ 3854, 3751, 3674, 3649, 3629, 3362, 3099, 2934, 2853, 2363, 1717, 1695, 1669, 1599, 1559, 1517, 1436, 1384, 1313, 1276, 1260, 1241, 1221, 1173, 1142, 1126, 1109, 1055, 1020, 959, 921, 889, 867, 827, 738, 691. ^1^H NMR (400 MHz, DMSO-*d_6_*) δ 8.36 (1H, s, H-8); 8.24 (2H, d, *J* = 10.2, CONH_2_); 7.85 (1H, s, H-5); 3.13–3.09 (4H, br. m, 2 × CH_2_); 2.41 (3H, s, CH_3_); 1.73–1.68 (4H, br. m, 2 × CH_2_); 1.62–1.57 (2H, br. m, CH_2_). ^13^C NMR (100 MHz, DMSO-*d_6_*) δ160.6(CO); 153.3 (6-C); 139.2 (3-C); 137.2 (9-C); 136.5 (10-C); 132.9 (7-CCl); 131.46 (2-C); 121.1 (8-CH); 108.1 (5-CH); 52.1 (2 × CH_2_); 25.5 (2 × CH_2_); 23.4 (CH_2_); 14.4 (CH_3_). HRMS (ESI) calculated for C_15_H_18_ClN_4_O_3_^+^ [M+H]^+^ 337.1062, found 337.1056.

#### 4.2.8. 6-Chloro-2-(ethoxycarbonyl)-3-methyl-7-(piperazin-1-yl)quinoxaline 1,4-dioxide hydrochloride (**7**)

This compound was isolated as a minor product in the synthesis of the derivative **4** described in Section 4.2.5. R_f_ = 0.62 (hexane–ethyl acetate, 3:1). It was a yellow powder (yield 21%, mp 205–207 °C). HPLC (LW = 370 nm, gradient B 10/50% (45 min)) t_R_ = 14.5 min, purity 94.7%. λ_max._, EtOH: 216, 274, 328, 371 nm. IR ν_max_, (film) cm^−1^ 3561, 3479, 3083, 3009, 2911, 2785, 2697, 2465, 2379, 1735, 1621, 1598, 1547, 1510, 1472, 1433, 1415, 1377, 1336, 1321, 1296, 1264, 1248, 1235, 1222, 1194, 1177, 1150, 1166, 1131, 1084, 1059, 1025, 982, 931, 904, 888, 871, 854, 839, 817, 781, 764, 733, 707. ^1^H NMR (500 MHz, DMSO-*d_6_*) δ 9.91 (2H, br. s, NH_2_^+^); 8.41 (1H, s, H-5); 7.82 (1H, s, H-8); 4.49 (2H, q, *J* = 7.0, O**CH_2_**CH_3_); 3.43–3.39 (4H, br. m, 2 × CH_2_); 3.30–3.27 (4H, br. m, 2 × CH_2_); 2.40 (3H, s, CH_3_); 1.35 (3H, t, *J* = 7.0, OCH_2_**CH_3_**). ^13^C NMR (125 MHz, DMSO-*d_6_*) δ159.5(CO); 150.3 (7-C); 137.8 (3-C); 135.8 (9-C); 135.2 (2-C); 133.9 (6-CCl); 133.7 (10-C); 121.1 (5-CH); 108.9 (8-CH); 63.3 (O**CH_2_**CH_3_); 47.6 (2 × CH_2_); 42.7 (2 × CH_2_); 13.9 (CH_3_); 13.8 (OCH_2_**CH_3_**). HRMS (ESI) calculated for C_16_H_20_ClN_4_O_4_^+^ [M+H]^+^ 367.1168, found 367.1169.

#### 4.2.9. 6,7-Dichloro-2-(ethoxycarbonyl)-3-trifluoromethylquinoxaline 1,4-dioxide (**8**)

The compound **8** was previously synthesized in [11]. It was a yellow powder (yield 46%, mp 172–173 °C). ^1^H NMR (400 MHz, DMSO-*d_6_*) δ 8.72 (1H, s, H-8); 8.68 (1H, s, H-5); 4.50 (2H, q, *J* = 7.0, O**CH_2_**CH_3_); 1.34 (3H, t, *J* = 7.0, OCH_2_**CH_3_**). ^13^C NMR (100 MHz, DMSO-*d_6_*) δ157.5(CO); 138.1 (2-C); 137.9 (9-C); 137.2 (6-C); 137.6 (7-C); 134.2 (10-C); 129.1 (q, *J* = 33.7, 3-C); 121.8 (5-CH); 121.5 (8-CH); 118.8 (q, *J* = 273.8, CF_3_); 63.9 (O**CH_2_**CH_3_); 13.6 (OCH_2_**CH_3_**). 

#### 4.2.10. 6-Chloro-2-(ethoxycarbonyl)-3-trifluoromethyl-7-(piperazin-1-yl)quinoxaline 1,4-dioxide hydrochloride (9)

The compound **9** was previously synthesized in [11]. It was a yellow powder (yield 83%, mp 253–254 °C (dec.)). HPLC (LW = 280 nm, gradient B 20/80% (455 min)) t_R_ = 13.9 min, purity 96.6%. λ_max._, EtOH: 235, 267, 282, 345, 393 nm. ^1^H NMR (400 MHz, DMSO-*d_6_*) δ8.97 (2H, br. s, NH_2_^+^); 8.50 (1H, s, H-8); 7.86 (1H, s, H-5); 4.47 (2H, q, *J* = 7.0, O**CH_2_**CH_3_); 4.31–4.29 (4H, br. m, 2 × CH_2_); 3.46–3.44 (4H, br. m, 2 × CH_2_); 1.32 (3H, q, *J* = 7.0, OCH_2_**CH_3_**). ^13^C NMR (100 MHz, DMSO-*d_6_*) δ 157.8 (CO); 152.8 (7-C); 138.0 (9-C); 134.6 (2-C); 134.4 (10-C); 133.8 (6-CCl); 127.3 (d, *J* = 33.7, 3-C); 121.9 (5-CH); 118.9 (q, *J* = 272.9, CF_3_); 108.6 (8-CH); 63.8 (O**CH_2_**CH_3_); 47.6 (2 × CH_2_); 42.9 (2 × CH_2_); 13.7 (OCH_2_**CH_3_**). HRMS (ESI) calculated for C_16_H_17_ClF_3_N_4_O_4_^+^ [M+H]^+^ 421.0885, found 421.0883.

### 4.3. Study of the Stability of Compounds **4**–**7**

The stability of compounds **4**–**7** was assessed using the accelerated aging method (storage at 37 °C), and the impurity content was determined over a period of 7 days. A sample weighing 5–10 mg of the substance was dissolved in a glass vial with distilled water, resulting in a concentration of 1 mg/mL. The vial was then sealed and placed in a thermostat at 37 °C for 7 days. After the thermostatting period, the contents of both the main substance and the impurities were analyzed using gradient HPLC.

### 4.4. Biology

#### 4.4.1. Microbial Cultures and Growth Conditions

All the bacterial cultures used in this study are summarized in Table 4.

*M. smegmatis* strains were grown in liquid Middlebrook 7H9 medium (Himedia, Mumbai, India) supplemented with oleic albumin dextrose catalase (OADC, Himedia), 0.1% Tween-80 (*v*/*v*), and 0.4% glycerol (*v*/*v*), while M290 Soyabean Casein Digest Agar (Himedia) was used as the solid media. *Escherichia coli* DH5α was used for plasmid propagation and was grown in liquid or on agarized LB medium with the addition of the corresponding selection antibiotics when required. Cultures in liquid medium were incubated in a Multitron incubator shaker (Infors HT, Bottmingen-Basel, Switzerland) at 37 °C and 250 rpm.

#### 4.4.2. Minimal Inhibitory Concentration Determinations

##### MIC Determination on *M. smegmatis* Strains

Minimal inhibitory concentrations (MICs) of the studied compounds on *M. smegmatis* were determined in a liquid medium. *M. smegmatis* strains were cultured overnight in 7H9 medium and then diluted at a proportion of 1:200 in fresh medium (to approximately OD_600_ = 0.05). Then, 196 μL of the diluted culture was poured in sterile non-treated 96-well flat-bottom culture plates (Eppendorf, Hamburg, Germany); 4 μL of serial two-fold dilutions of the tested compounds in DMSO were added to the wells to reach final concentrations ranging from 0.5 to 32 μg/mL, while 4 μL of DMSO was used as an untreated control. The plates were incubated at 37 °C and 250 rpm for 48 h. The experiments were carried out in triplicates, and the MIC was determined as the lowest concentration of the compound with no visible bacterial growth.

##### MIC Determination on *M. tuberculosis* Strains 

Serial dilutions of drug-containing solutions and autoluminescent *M. tuberculosis* H37Ra (AlRa) strain broth culture (OD_600_ of 0.3 to 1.0) were prepared. Relative light unit (RLU) counts from the same batch of triplicate samples were measured according to the designed time points. Ninety-six-well plates were measured as rendering the amount of light by using a VeritasTM Microplate Luminometer Operating Manual (Promega, Madison, WI, USA). The MIC_lux_ was determined as the lowest concentration that could inhibit > 90% of the RLUs compared with that of the untreated controls. The MIC_lux_ of the autoluminescent strains was determined by detecting RLUs from these samples as described previously [26,27]. The concentrations used were a two-fold series of dilutions for this purpose that ranged from 0.078 to 20 μg/mL (a total of nine concentrations). The light from each 1.5 mL tube containing a total of 500 μL of culture was detected using GloMax 20/20n (Promega, Madison, WI, USA). Each experiment above was performed in triplicate (three independent experiments) or more by independent persons and from three or more independent cultures.

#### 4.4.3. Paper Disk Assay

*M. smegmatis* cultures were incubated at 250 rpm and 37 °C overnight until OD_600_ = 1.2. Afterwards, the drug susceptibility was assessed by using a paper-disc assay: *M. smegmatis* culture was diluted 1:9:10 (culture:water:М290 medium) and seeded over the base agar layer on Petri dishes. The compounds were manually impregnated on sterile paper discs. The plates were incubated for 2 days at 37 °C until the bacterial lawn was fully grown. Growth inhibition halos were measured to the nearest 0.1 mm (the halo area around the disk was measured with a digital caliper and analyzed in a Mega Bio-Print 3020-WL/LC/20M X-Press, Vilber Lourmat Gel Documentation System). The experiments were carried out as three repeats, and the average diameter and standard deviation (SD) were calculated. We used CLSI standards containing information about disk diffusion (M02) and dilution (M07, M11) test procedures for bacteria and Laboratory Practices in Microbiology [28]. 

#### 4.4.4. Generation of Resistant Mutants and Their Phenotype Characterizations

The *M. smegmatis* strain was grown in a liquid medium to reach OD_600_ = 2.8 (~4 × 10^8^ CFU/mL). Then, 200 μL of bacterial culture was plated on agar plates containing **4** at a final concentration of 32 μg/mL. The plates were incubated in a thermostat at 30 °C for three days until the emergence of single colonies. The colonies from two plates for each strain were counted to determine the frequency of drug-resistant mutants’ emergence. Serial 10-fold dilutions of each bacterial culture were plated on compound-free plates to determine the exact titer. The mutants’ resistance phenotypes were confirmed by streaking several colonies on M290 plates containing 32 μg/mL of **4**. The w.t. strain was used as a control. The MICs on resistant mutants were determined in a liquid medium as described above. 

#### 4.4.5. *M. smegmatis* Whole-Genomic Sequencing and Analysis 

Mycobacterial genomic DNA was isolated from 10 mL via enzymatic lysis as described by Belisle et al. [29]. After preliminary isolation, the DNA was treated with RNase A (Thermo Fischer Scientific, Waltham, MA, USA) and extracted in the phenol–chloroform–isoamyl alcohol solution (25:24:1). 

A total of 250 ng of genomic DNA was taken for shotgun sequencing library preparation. After DNA sonication on a Covaris S220 System (Covaris, Woburn, MA, USA), the size (400–500 b.p.) and quality of the fragmented samples were assessed on an Agilent 2100 Bioanalyzer (Agilent, Santa Clara, CA, USA) according to the manufacturer’s manual. A NEBNext Ultra II DNA Library Prep Kit (New England Biolabs, Ipswich, MA, USA) was used for pair-ended library preparation, and a NEBNext Multiplex Oligos kit for Illumina (96 index primers, New England Biolabs) was used for library indexing. The libraries were quantified with a Quant-iT DNA Assay Kit, High Sensitivity (Thermo Fischer Scientific). DNA sequencing (2 × 125 b.p.) was performed on a HiSeq 2500 platform (Illumina, San Diego, CA, USA) according to the manufacturer’s recommendations. 

The quality of the obtained reads was assessed with FastQC (v. 0.11.8) (FastQC: A Quality Control Tool for High Throughput Sequence Data); as the quality was good, we proceeded to the assembly without trimming the reads. The reads were aligned to the reference genome (NC_008596.1, PRJNA57701) using the BWA-MEM algorithm [30]. The pileup was generated with mpileup (-B-f) in SAMtools [31]. Single-nucleotide variants were called by running mpileup2snp (–min-avg-qual 30–min-var-freq 0.80–*p*-value 0.01–output-vcf 1–variants 1) in VarScan (v. 2.3.9) [32]. The annotation was created using vcf_annotate.pl (developed by Natalya Mikheecheva of the Laboratory of Bacterial Genetics, Vavilov Institute of General Genetics, Moscow, Russia). The non-synonymous single-nucleotide variants found within open reading frames and absent in the wild-type strain were selected for further analysis. The raw sequencing data (SRA) are publicly available in NCBI GenBank (BioProject ID: PRJNA778794).

#### 4.4.6. Construction of the *M. smegmatis* Recombinant Strains 

To generate *M. smegmatis* strains carrying single mutations in the genes *MSMEG_4646*, *MSMEG_4648*, and *MSMEG_5122*, based on the *M. smegmatis* mc^2^ 155 strain, homologous recombination using the suicide system p2NIL/pGOAL19 [20] was applied. Briefly, using the Q5 high-fidelity PCR kit (NEB, Ipswich, MA, USA), primers containing specific restriction sites, genomic DNA from spontaneous *M. smegmatis* mutants with target mutations, corresponding fragments containing mutations in the target genes, and ~1500 bp homologous arms for homologous recombination were amplified. After digestion with appropriate restriction endonucleases (Thermo Fisher Scientific, Waltham, MA, USA) of the amplicons and the p2NIL plasmid, ligation and selection of the target clones were performed using the method described above. Next, the target plasmids were cloned with a cassette containing marker genes from the pGOAL19 plasmid at the *Pac*I restriction site. The final plasmids were electroporated into *M. smegmatis* mc^2^ 155 cells. Selection of single crossovers was performed on tryptone–soy agar containing kanamycin (50 μg/mL), hygromycin (50 μg/mL), and X-Gal (50 μg/mL). Since the p2NIL plasmid lacks mycobacterial replication origin, the function of marker genes was only possible upon plasmid integration into the chromosome via homologous recombination. Blue single-crossover colonies were grown overnight in liquid Middlebrook 7H9 medium and then plated onto tryptone–soy agar containing X-Gal (50 μg/mL) and 10% sucrose for selection of double crossovers, and their sensitivity to kanamycin was tested to confirm the complete removal of the cassette containing marker genes from the chromosome. For final selection of recombinants carrying the desired single-nucleotide replacement, the target region was sequenced via Sanger sequencing.

#### 4.4.7. RNA Extraction and Real-Time PCR

To isolate the total RNA, *M. smegmatis* cultures were grown in 10 mL Middlebrook 7H9 medium until reaching an optical density (OD_600_) of approximately 1.0, after which they were washed twice with 3 mL of RNAprotect Bacteria Reagent (QIAGEN, Germantown, MD, USA). The total bacterial RNA was extracted using the RNeasy Plus Mini Kit (QIAGEN, USA) following the manufacturer’s protocol with the use of Lysing Matrix B silicon beads (MP Biomedicals, Santa Ana, CA, USA) for homogenization. After extraction and purification, the RNA was treated with TURBO DNAse (Thermo Fisher Scientific, USA), and the concentration was measured using the Qubit instrument (Invitrogen, Waltham, MA, USA) with an RNA BR Assay Kit (Thermo Fisher Scientific, USA). The integrity of the RNA was evaluated via agarose gel electrophoresis. The first-strand cDNA synthesis was performed using the iScript Select cDNA Synthesis Kit (Bio-Rad, USA) with 300 ng of total RNA.

For quantitative real-time PCR, the qPCR-HS SYBR Kit (Eurogen, Moscow, Russia) and 5 ng of cDNA were used on the CFX96 Touch instrument (Bio-Rad, Hercules, CA, USA). The results were analyzed using CFX Manager V 3.1 software (Bio-Rad, USA), and the relative normalized expression was calculated as ∆∆Cq using the *sig*A and *fts*Z genes as references.

#### 4.4.8. Cytotoxic Measurement Assay 

The cytotoxic effects of various concentrations of derivative **4** on human dermal fibroblasts (HDF) were measured using an MTT assay. This assay is based on the measurement of succinate dehydrogenase activity, a mitochondrial enzyme, and is widely used for in vitro toxicity testing of potential drugs. Using the MTT assay data, the half-maximum inhibition concentration (IC_50_) was calculated. The primary HDF cells were obtained from a healthy donor. The HDF cells were cultured in DMEM medium (10% FBS, 2 mM glutamine, and 1% gentamicin) in plastic flasks under sterile conditions and incubated at 37 °C in the presence of 5% CO_2_. A stock solution of **4** (50 mM) was prepared in DMSO; before adding it to the cells, it was diluted to the desired concentrations in the culture medium, and HDF cells were seeded into the wells of 96-well plates at a density of 3.5 × 10^3^ cells per well. Cells were allowed to attach for 14 h, after which different concentrations of the test compound or DMSO (as a control sample) were injected in triplicate using the titration method. The final volume in the wells was 100 μL. After 48 h of sample addition, cell viability was measured using the MTT reagent (Sigma, Livonia, MI, USA) at a volume of 10 μL per well. In each well, the working solution (7 mg/mL) was added (100 μL per well), and the wells were incubated for 3 h, after which the medium was replaced with DMSO solution. The optical density of each well was determined at 570 nm using a plate spectrophotometer (TECAN Infinite M Plex) with subtraction of the background absorbance. The concentration value and the inhibitory dose (IC_50_) were determined from the dose-response curves.

#### 4.4.9. In Vivo Activity against *M. tuberculosis*

A rapid drug susceptibility screening of the compound was carried out in vivo using AlRa. Briefly, at Day -1 (D -1), each of the BALB/c mice was intravenously infected with 250 µL of a mid-log phase culture of AlRa at ~2,000,000 relative light units per milliliter (RLU/mL). In addition, each mouse was identified with a color tag for individual follow-up of the drug effect on bacterial growth. Compound **4** was dissolved in 10% Tween 80, and solvent and rifampicin were used as negative and positive controls, respectively. Daily treatment was initiated a day after infection (Day 0 (D 0)), and bacterial growth was monitored at two-day intervals and on the last day of treatment (Day 5 (D 5)). Drugs were administered through the following routes at the following doses (mg/kg): rifampicin, 10 (oral); **4,** 30 (oral); **4,** 100 (oral); **4,** 100 (intraperitoneally); and **4,** 200 (oral).

## 5. Conclusions

We evaluated newly synthesized quinoxaline-1,4-dioxide derivatives for their antimicrobial efficacy against mycobacterial strains. Of these, derivative 4 emerged as the most potent candidate with MICs of 1.25 and 4 μg/mL against *M. smegmatis* and *M. tuberculosis*, respectively. Our investigations suggest its mechanism of action centers on inducing bacterial DNA damage and interfering with pyruvate metabolism. Significantly, in vivo studies highlighted its low toxicity profile. Our future endeavors aim to produce more potent derivatives of this compound while preserving its low toxicity. In essence, derivatives of quinoxaline-2-carboxylic acid 1,4-dioxide stand out as a promising foundation for the formulation of advanced antimycobacterial agents.

## Data Availability

Data is contained within the article and Appendix A.

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
