# Peer review of "Novel Derivatives of Quinoxaline-2-carboxylic Acid 1,4-Dioxides as Antimycobacterial Agents: Mechanistic Studies and Therapeutic Potential"

_pharmaceuticals, 2023, doi:10.3390/ph16111565_

Round 1

Reviewer 1 Report

Comments and Suggestions for Authors

The manuscript reports the synthesis, characterization and biological evaluation of novel quinoxaline-2-carboxylic acid 1,4-dioxide derivatives as potential antimycobacterial agents. The manuscript is well written and can be accepted for publication after consideration of the following issues:

1) Abstract should be rewritten to point out the main findings of the research.

2) Spectroscopic data for the novel compounds should be included in Results and Discussion sections.

3) Solution stability study for the tested compounds should be performed (e.g. by 1H NMR spectroscopy).

4) MIC values for the tested compounds should be also given in micromolar concentrations.

Author Response

Dear Reviewer,

We would like to thank you for the constructive review of our manuscript. The provided comments are very helpful and we have revised the manuscript as suggested. Please find below the point-by-point response to the reviewer's comments.

“1. Abstract should be rewritten to point out the main findings of the research.”

Response: The abstract has been rewritten in accordance with the comments. (lines 38).

“2. Spectroscopic data for the novel compounds should be included in Results and Discussion sections.

Response: Thank you for your suggestions. The spectroscopic data of novel compounds were added in Results and Discussion sections.

“3. Solution stability study for the tested compounds should be performed (e.g. by 1H NMR spectroscopy).”

Response: Thank you for your suggestion. The solution stability data for described compounds were in the text.

“4. MIC values for the tested compounds should be also given in micromolar concentrations”

Response: MICs values in micromolar concentrations were added into tables 1 and 3.

We have also made some minor style and typos corrections, and reformatted the

references section to include the references added during the revision. I hope the revised manuscript is now acceptable for publication. Thank you for your consideration.

Best wishes,

Svetlana G. Frolova,  Researcher,

Laboratory of Bacterial Genetics,

Vavilov Institute of General Genetics RAS, 3 Gubkina str., 119333, Moscow, Russia.

Mobile: +7 9260524354

Work: +7 (499) 135-12-39

Reviewer 2 Report

Comments and Suggestions for Authors

The work describes the synthesis and antimycobacterial activities of nine quinoxaline derivatives. The manuscript is written nicely, however minor improvement is required. 

1. Abstract should include the synthesis of newer compounds.

2. The spectral data should include the IR spectra of all the newly synthesized compounds.

3. Compounds 8 and 9 spectral data is missing.

4. As compounds 8 and 9 contains triflouromethyl group, therefore  CF3 splitting observed in the 13C NMR should also be discussed.

5. Molecular docking analysis of the active compounds will also provide better insight into their molecular mechanism.

6. The manuscript contains several typographical errors.

Comments on the Quality of English Language

The manuscript is written nicely, however requires minor editing for the typos and English grammar.

Author Response

Dear Reviewer,

We would like to thank you for the constructive review of our manuscript. The provided comments are very helpful and we have revised the manuscript as suggested. Please find below the point-by-point response to the reviewer's comments.

“1. Abstract should include the synthesis of newer compounds.”

Response: The abstract has been rewritten in accordance with the comments. (lines 38).

“2. The spectral data should include the IR spectra of all the newly synthesized compounds.”

Response: The IR spectra were included in the Experimental section of the text of the Article and Supplementary Materials.

“3. Compounds 8 and 9 spectral data is missing.”

Response: The spectral data of the previously obtained and described compounds 8, 9 were added in the text of the Article and Supplementary Materials.

“4. As compounds 8 and 9 contains triflouromethyl group, therefore  CF3 splitting observed in the 13C NMR should also be discussed.”

Response: The confirmation of the structure of the 3-trifluoromethylderivatives 8 and 9 have been discussed in the Results and Discussion section.

“5. Molecular docking analysis of the active compounds will also provide better insight into their molecular mechanism”

Response: Thank you for this suggestion. Based on the mutations we have identified, one of the possible mechanisms of action of quinoxaline-1,4-dioxide derivatives may involve direct introduction of single- and double-stranded breaks in bacterial DNA, which could explain the multiple mutations observed in M. smegmatis qdr strains [21]. We have detected four mutations in the genes MSMEG_1380, MSMEG_4646, MSMEG_4648, and MSMEG_5122, which were found simultaneously in several qdr mutants, suggesting that the MmpS5-MmpL5 pump may play a role in antibiotic efflux from the cell. Therefore, for docking studies, we cannot definitively establish the target within the cell, as the primary proposed mechanism of action of these compounds involves inducing breaks in DNA.

“6. The manuscript contains several typographical errors.”

Response: Thank you for your comment, we checked the text of the article and tried to correct all typographical errors.

We have also made some minor style and typos corrections, and reformatted the references section to include the references added during the revision. I hope the revised manuscript is now acceptable for publication. Thank you for your consideration.

Best wishes,

Svetlana G. Frolova,  Researcher,

Laboratory of Bacterial Genetics,

Vavilov Institute of General Genetics RAS, 3 Gubkina str., 119333, Moscow, Russia.

Mobile: +7 9260524354

Work: +7 (499) 135-12-39

Reviewer 3 Report

Comments and Suggestions for Authors

In this manuscript, Frolova and coworkers designed and synthesized several quinoxaline-2-carboxylic acids 1,4-dioxides and tested their antimycobacterial activities. Considering the importance of these results and usefulness of it in future drug discovery studies, this manuscript is recommended for publication in Pharmaceuticals after addressing the following comments.

1.     Compound 4-7 are not carboxylic acid; they are ester and amide. Hence, author should remove “carboxylic acid” term from the title, abstract, keywords and main text while mentioning compound 4-7 and 9.

2.     Author should provide HRMS, ESI, HPLC, and NMR spectrum of Compound 4-7 and 9 in the revised version.

3.     How authors confirmed the structure of compound 4-7? They should provide 2D NMR correlation diagram of compounds 4-7 confirming the structure. What is the rationale for regioselectivity of amine addition?

4.     In Scheme 1, author indicated that SnAr reaction is preferred on the 6-position and isolated 6-amino substituted compounds 4-6. However, in scheme 2, author showed SnAr reaction on 7-position and isolated compound 9. They should explain why they observed opposite regioselectivity between those two cases.

Author Response

Dear Reviewer,

We would like to thank you for the constructive review of our manuscript. The provided comments are very helpful and we have revised the manuscript as suggested. Please find below the point-by-point response to the reviewer's comments.

“1. Compound 4-7 are not carboxylic acid; they are ester and amide. Hence, author should remove “carboxylic acid” term from the title, abstract, keywords and main text while mentioning compound 4-7 and 9.”

Response: Thank you for this remark. The described compounds are in fact ester and amide, rather than carboxylic acid. However they are all derivatives of the carboxylic acid, and we tried to emphasize it throughout the manuscript, by calling them “derivatives of … carboxylic acid”.

“2. Author should provide HRMS, ESI, HPLC, and NMR spectrum of Compound 4-7 and 9 in the revised version.”

Response:  We apology for these mistakes: in the initial files, all HRMS, ESI, HPLC, and NMR spectra of the obtained novel compounds in Experimental section as well as Supplementary Material were included, but after uploading the files, they were lost.

“3. How authors confirmed the structure of compound 4-7? They should provide 2D NMR correlation diagram of compounds 4-7 confirming the structure. What is the rationale for regioselectivity of amine addition?”

Response: Explanation of the regioselectivity of nucleophilic aromatic substitution of the halogen atom in the quinoxaline derivatives was added to the Discussion section.

“4. In Scheme 1, author indicated that SnAr reaction is preferred on the 6-position and isolated 6-amino substituted compounds 4-6. However, in scheme 2, author showed SnAr reaction on 7-position and isolated compound 9. They should explain why they observed opposite regioselectivity between those two cases.”

Response: The observed highest reactivity of the chlorine atom at position 7 of quinoxaline 1,4-dioxide 8 in comparison to its bioisosteric analogues 2, 3 in the substitution reaction could be explained with a highest electron-withdrawal character of the trifluoromethyl group at position 3 in relation to ethoxycarbonyl group at position 2 of the heterocycle of the derivative 8. This explanation for the regioselective substitution of halogen atom at position 7 in the 6,7-dihalo-3-trifluoromethyl derivative 8 was given in a previous paper where such derivatives were described.

We have also made some minor style and typos corrections, and reformatted the

references section to include the references added during the revision. I hope the revised manuscript is now acceptable for publication. Thank you for your consideration.

Best wishes,

Svetlana G. Frolova,  Researcher,

Laboratory of Bacterial Genetics,

Vavilov Institute of General Genetics RAS, 3 Gubkina str., 119333, Moscow, Russia.

Mobile: +7 9260524354

Work: +7 (499) 135-12-39

Round 2

Reviewer 3 Report

Comments and Suggestions for Authors

Author addressed all reviewer's comments and significantly improved the quality of the manuscript. Hence, it should be accepted for publication.